# Efficient Separation of Phytochemicals from *Muehlenbeckia volcanica* (Benth.) Endl. by Polarity-Stepwise Elution Counter-Current Chromatography and Their Antioxidant, Antiglycation, and Aldose Reductase Inhibition Potentials

**DOI:** 10.3390/molecules26010224

**Published:** 2021-01-04

**Authors:** Guang-Lei Zuo, Hyun Yong Kim, Yanymee N. Guillen Quispe, Zhi-Qiang Wang, Seung Hwan Hwang, Kyong-Oh Shin, Soon Sung Lim

**Affiliations:** 1Department of Food Science and Nutrition, Hallym University, 1 Hallymdeahak-gil, Chuncheon 24252, Korea; B16504@hallym.ac.kr (G.-L.Z.); 41310@hallym.ac.kr (H.Y.K.); yany24@snu.ac.kr (Y.N.G.Q.); wangzq2017@hbu.edu.cn (Z.-Q.W.); hsh@hallym.ac.kr (S.H.H.); tlsruddhek@hallym.ac.kr (K.-O.S.); 2Department of Molecular Medicine and Biopharmaceutical Sciences, Graduate School of Convergence Science and Technology, Seoul National University, Seoul 151742, Korea; 3Tumor Microenvironment Global Core Research Center, College of Pharmacy, Seoul National University, Seoul 151742, Korea; 4College of Public Health, Hebei University, Baoding 071002, China; 5R&D Center, Huons Co., Ltd., 55 Hanyangdaehak-ro, Ansan 15588, Korea; 6Institute of Korean Nutrition, Hallym University, 1 Hallymdeahak-gil, Chuncheon 24252, Korea; 7Institute of Natural Medicine, Hallym University, 1 Hallymdeahak-gil, Chuncheon 24252, Korea

**Keywords:** antioxidant, antiglycation, aldose reductase inhibition, counter-current chromatography, *Muehlenbeckia volcanica* (Benth.) Endl., stepwise elution

## Abstract

*Muehlenbeckia volcanica* (Benth.) Endl. (*M. volcanica*), native to South America, is a traditional Peruvian medicinal plant that has multi-therapeutic properties; however, no phytochemicals have been identified from it yet. In this study, a five-step polarity-stepwise elution counter-current chromatography (CCC) was developed using methanol/water (1:5, *v*/*v*) as the stationary phase and different ratios of *n*-hexane, ethyl acetate, and *n*-butanol as mobile phases to separate the compounds from the 70% methanol extract of *M. volcanica,* by which six compounds with a wide range of polarities were separated in a single run of CCC and were identified as gallic acid, protocatechuic acid, 4,4′-dihydroxy-3,3′-imino-di-benzoic acid, rutin, quercitrin, and quercetin. Then, two compounds from the fractions of stepwise elution CCC were separated using conventional high-speed CCC, pH-zone-refining CCC, and preparative high-performance liquid chromatography, and identified as shikimic acid and miquelianin. These compounds are reported from *M. volcanica* for the first time. Notably, except for shikimic acid, all other compounds showed anti-diabetic potentials via antioxidant, antiglycation, and aldose reductase inhibition. The results suggest that the polarity-stepwise elution CCC can be used to efficiently separate or fractionate compounds with a wide range of polarities from natural products. Moreover, *M. volcanica* and its bioactive compounds are potent anti-diabetic agents.

## 1. Introduction

Targeting antioxidants, advanced glycation end products (AGEs), and aldose reductase (AR) as potential therapeutic approaches for the prevention and amelioration of diabetic complications has attracted growing interest [1,2,3]. An overload of glucose induces the overproduction of reactive oxygen species (ROS) in mitochondrial, and the rise of ROS alters the mitochondrial itself and the endoplasmic reticulum, thereby further increasing oxidative stress, which is a key contributor of diabetic complications [3]. Particularly, in hyperglycemia conditions, an important diabetic characteristic [4], the polyol pathway flux is significant in insulin-independent tissues, such as lens, glomerulus, and neural tissue, and it may cause a decrease in cytosolic NADPH/NADP^+^ and an increase in cytosolic NADH/NAD^+^ leading to ROS production and oxidative stress in those tissues. Moreover, the activated polyol pathway may also cause sorbitol-induced osmotic stress and fructose overproduction [4,5]. However, the elevated fructose and its derivative fructose-3-phosphate may react with amino groups in proteins to AGEs and contribute to diabetic complications [6,7]. Since ROS, AGEs, and the sorbitol-induced osmotic stress play an important role in the development of diabetic complications, such as diabetic neuropathy, retinopathy, and nephropathy [8], antioxidants, antiglycation agents, and inhibitors of AR, a key enzyme in the polyol pathway, have therapeutic potential for diabetic complications [1,2,3].

Natural products are important resources of anti-diabetic agents, among which medicinal plants play an important role. The use of medicinal plants to treat diabetes and its complications has a long history, and it is still an important alternative medicine therapy for the treatment of diabetic complications and has been recommended by the World Health Organization [9,10]. *Muehlenbeckia volcanica* (Benth.) Endl. (*M. volcanica*) is an herbaceous plant of the family Polygonaceae, native to South America [11]. It is traditionally used as a medicinal plant in Peru to treat kidney and liver inflammations, headache, arthritis, bone pain, rheumatism, and cancer [12,13]. Recently, it has been reported to have anti-radicals, AR inhibition, and hypoglycemic activities [14,15], and it is now receiving increasing attention [16]; however, the phytochemicals it contains, especially the bioactive compounds, are still unknown [16]. In addition, no studies have linked it so far with antiglycation effects. The anti-diabetic potential of *M. volcanica* promotes us to separate and identify the bioactive compounds it contains and to evaluate their antioxidant, antiglycation, and AR inhibitory potentials.

Possessing advantageous properties of high separation efficiency, high sample loading capacity, and absence of irreversible adsorption, etc., high-speed counter-current chromatography (HSCCC) has been widely used for the separation of bioactive compounds from natural products [17,18]. However, the separation of multiple compounds with a wide range of polarities from a complex natural product could not be satisfactorily achieved by using a single solvent system. In contrast, the stepwise elution counter-current chromatography (CCC) approach is more powerful to separate the compounds with a wide range of polarities from natural product extracts, which can offer a broader range of polarities to be covered in one run than in isocratic mode, and, therefore, improves the separation efficiency and minimizes the use of solvents and time [19,20,21,22]. In the present study, a five-step solvent polarity-gradient elution CCC using methanol/water (1:5, *v*/*v*) as the stationary phase and different ratios of *n*-hexane, ethyl acetate, and *n*-butanol as mobile phases was developed and used to separate the components from the 70% methanol extract of *M. volcanica*. In addition, conventional HSCCC, pH-zone-refining CCC, a technique for the separation of ionizable compounds [23], and preparative high-performance liquid chromatography (HPLC) were also used for the separation and purification of two sub-fractions of the polarity-stepwise elution CCC.

Therefore, we aim to develop a polarity-stepwise elution CCC to efficiently separate or fractionate the phytochemicals from the 70% methanol extract of *M. volcanica* and to characterize their structures and antioxidant, antiglycation, and aldose reductase inhibitory activities.

## 2. Results and Discussion

### 2.1. Antioxidant, Antiglycation, and Aldose Reductase Inhibitory Activities of the 70% Methanol Extract of M. volcanica

Targeting antioxidants, AGEs, and AR as potential therapeutic approaches for the prevention and amelioration of diabetic complications have attracted growing interest [1,2,3], whereas natural products play an important role [9,10]. In this study, the anti-diabetic potential of the 70% methanol extract of *M. volcanica* was evaluated via its capacities of antioxidant, antiglycation, and AR inhibition.

DPPH (2,2-diphenyl-1-picrylhydrazyl) and ABTS (2,2′-Azino-bis(3-ethylbenzothiazoline-6-sulfonic acid) diammonium salt) assays have been widely used to determine the antioxidant activities of natural products as they are simple to operate, rapid to obtain results, and low-cost, whereas ORAC (oxygen radical absorbance capacity) assay combines both inhibition percentage and time of the free radical action by antioxidants in a single quantity [24]. Since no single method is sufficient to evaluate the antioxidant activity of samples, therefore, these three methods were performed in this study. The results were expressed as Trolox equivalent antioxidant capacity (TEAC, µg Trolox/µg extract), which were calculated with the equations obtained from the calibration curves of Trolox standards (final concentrations) versus their DPPH and ABTS radical inhibition (%) and the net area under the curve (AUC) value (ORAC assay). A larger TEAC value indicates that the sample tested has higher antioxidant activity. As shown in Table 1, a strong radical scavenging activity against DPPH and ABTS radicals was found in the 70% methanol extract of *M. volcanica*, which exhibited about half the antioxidant activity of Trolox (0.48–0.49 µg Trolox/µg extract), and this is consistent with a previous study (documented as synonym *Muehlenbeckia volcanica* Meisn.) [14]. Moreover, for the first time, the extract was found to have higher peroxyl radical scavenging (ORAC assay) activity than Trolox by showing a TEAC value of 1.10 µg Trolox/µg extract.

The antiglycative activity of the extract was determined using fructose-mediated bovine serum albumin (BSA) glycation assay, as described in Section 3.4. Both the extract and the positive control aminoguanidine hydrochloride, a synthetic antiglycation agent [25], showed a dose-dependent antiglycation effect (Figure 1A). Notably, the extract was much more active than aminoguanidine hydrochloride against protein glycation by showing their antiglycation IC_50_ values of 21.35 µg/mL and 317.36 µg/mL, respectively.

As shown in Figure 1B, both quercetin, a known AR inhibitor [26], and the 70% methanol extract of *M. volcanica* showed good dose-dependent AR inhibitory activities and, especially, the latter showed higher AR inhibitory activity than the former by giving IC_50_ values of 5.37 µg/mL and 1.18 µg/mL, respectively.

Possessing remarkable antioxidant, antiglycation, and AR inhibitory activities, the 70% methanol extract of *M. volcanica* is a promising anti-diabetic agent, which encourages us to separate and identify the bioactive compounds from this extract.

### 2.2. Separation of Compounds by Polarity-Stepwise Elution CCC

As a versatile separation technique based on fast solvent-solvent partition, HSCCC has many advantageous properties such as high separation efficiency, high sample loading capacity, absence of irreversible absorption, and low risk of sample denaturation, and, therefore, it has been widely used for the separation of bioactive compounds from natural products [17,18]. A successful separation by conventional HSCCC usually relies on the selection of a suitable biphasic solvent system that offers ideal partition coefficients (*K* values) (0.5 ≤ *K* ≤ 2.0) [17] or “sweet spot” *K* values (0.4 ≤ *K* ≤ 2.5) [27], and suitable separation factors (*α* values) (*α* ≥ 1.5) [16]. We first tested a series of isocratic *n*-hexane/ethyl acetate/*n*-butanol/methanol/water solvent systems developed by Ito [23] to determine the *K* values of the compounds in the extract. As shown in Appendix A, the compounds from the 70% methanol extract of *M. volcanica* were found to have a wide range of polarities; however, more than six runs of conventional HSCCC could be required for the separation or fractionation of those compounds, which would be time- and solvent-consuming. Therefore, a more efficient solvent polarity-stepwise elution CCC separation strategy was proposed.

The methanol/water (1:5, *v*/*v*) solvent was selected as the stationary phase of the polarity-stepwise elution CCC, since the settling times and phase ratios (upper phase to lower phase) of the newly paired solvent systems composed of the stationary phase, methanol/water (1:5, *v*/*v*), and individual mobile phases with polarities from low to high (Table 2), were within the range of the recommended values: their settling times were less than 30 s, and the phase ratios were close to 1 (Table 2). A short settling time usually facilitates the retention of the stationary phase in the HSCCC column and a rapid settling time is recommended to be shorter than 30 s [17]. That the phase ratios of each newly paired solvent system were close to 1 might help maintain the retention of the stationary phase. Thus, all the solvents shown in Table 2 are acceptable as alternative mobile phases for the stepwise elution CCC separation.

Next, the *K* values of the compounds in the extract were measured using the newly paired solvent systems, and the data are shown in Table 2. The *K* values offered by the selected solvent systems were essential for a successful CCC separation, and a suitable *K* value is recommended to be 0.5 < *K* < 2.0 [17]. Accordingly, a total of five mobile phases were selected for the polarity-stepwise elution CCC as follows: mobile phase I, *n*-hexane/ethyl acetate (3:5, *v*/*v*), for compound **10** (*K*_10_ = 12.21); mobile phase II, *n*-hexane/ethyl acetate (2:5, *v*/*v*), for compounds **4** and **6** (*K*_4_ = 0.88, *K*_6_ = 1.49); mobile phase III, *n*-hexane/ethyl acetate (1:5, *v*/*v*), for compounds **5** and **9** (*K*_5_ = 0.86, *K*_9_ = 1.13); mobile phase IV, ethyl acetate, for compound **3** (*K*_3_ = 1.38); and mobile phase V, ethyl acetate/*n*-butanol (4:1, *v*/*v*), for compounds **7** and **8** (*K*_7_ = 0.80, *K*_8_ = 1.17).

Then, the separation was carried out as described in Section 3.7.2. As shown in Figure 2, compounds **10** (13.4 mg), **6** (18.0 mg), **4** (55.1 mg), **9** (57.8 mg), **3** (27.7 mg), and **8** (42.1 mg), with polarities from low to high, were sequentially separated from the 70% methanol extract of *M. volcanica* (2 g, Figure 3A) by polarity-stepwise elution CCC using mobile phases I to V. The purity of the separated compounds was determined to be 95.8%, 98.9%, 99.7%, 95.1%, 98.4%, and 94.2% for compounds **3**, **4**, **6**, **8**, **9**, and **10**, respectively, using HPLC at 250 nm (Figure 3). Compounds **1** and **2** were retained in the stationary phase due to their high polarity, which were collected and evaporated (1.320 g, Figure 3B). Moreover, an enrichment of compound **5** (56.6 mg, Figure 3E) and compound **7** (172.0 mg, Figure 3G) was also achieved by polarity-stepwise elution CCC. The final stationary phase retention ratio was measured to be about 25%. In this stepwise elution CCC separation, mobile phases II to V were individually prepared in glass bottles without mixing with the stationary phase prior to use, thereby reducing the solvents used. However, using the mobile phases that have not previously been saturated (or mixed) with the stationary phase may change the composition of the stationary phase, resulting in the loss of the stationary phase, as happened in this study (the final stationary phase retention ratio was about 25%). Therefore, future CCC separation using this stepwise elution CCC solvent systems is suggested to saturate the mobile phases prior to use by a suitable amount of stationary phase as previously described [28].

In a single run of polarity-stepwise elution CCC, six compounds with a wide range of polarities were directly separated; however, due to the big difference of the *K* values appeared in these compounds caused by their wide range of polarities, a multi-run of conventional HSCCC would be required for the separation or fractionation of these compounds, which would be quite laborious and time- and solvent-consuming. Nevertheless, the components from natural products, e.g., plant resources, are typically complex and carry a wide range of polarities, and therefore, the separation or fractionation of these compounds is usually difficult. Our result suggested that the polarity-stepwise elution CCC holds the potential to be used for the efficient separation or fractionation of the components with a wide range of polarities from natural products.

### 2.3. Purification of Compound ***2*** by Conventional HSCCC and Preparative HPLC

As shown in Figure 3B, compounds **1** and **2**, concentrated by stepwise elution CCC, could not be easily separated by column chromatography due to their close HPLC retention times, and then, the conventional HSCCC was used to separate compounds **1** and **2**. As shown in Table 3, four solvent systems were tested, and 26 mM of the trifluoroacetic acid-modified solvent system *n*-butanol/water (1:1, *v*/*v*) was selected to separate compounds **1** and **2** by conventional HSCCC, since the *K* and *α* values of compounds **1** and **2** offered by this solvent system were acceptable by giving *K*_1_ = 0.59, *K*_2_ = 0.39, and *α* (*K*_1_/*K*_2_) = 1.52. Then, the HSCCC separation (Figure 4A) was carried out using this solvent system as described in Section 3.8.1. About 732 mg of compound **2** (Figure 4D) was separated from 1 g of the mixture of **1** and **2** (Figure 4C). Interestingly, compound **1** could not be found in any of the HSCCC fractions or in the solvent remaining in the coiled column. Then, about 190 mg of compound **2** (Figure 4D) was further purified using preparative HPLC (Figure 4B), and about 152.8 mg of high-purity compound **2** (Figure 4E) was obtained.

### 2.4. Purification of Compound ***7*** by pH-Zone-Refining CCC and Preparative HPLC

As shown in Figure 5C (or Figure 3G), compound **7** was concentrated by stepwise elution CCC as a mixture of compounds **7** and **8**, however, they might be difficult to separate using column chromatography due to their close HPLC retention times. So, we decided to separate compound **7** using HSCCC. Considering that the α value between compounds **7** and **8**, offered by the solvent system (ethyl acetate/*n*-butanol 4:1, *v*/*v*)/(methanol/water 1:5, *v*/*v*) 1:1, *v*/*v*, was less than 1.5 (Table 2, *K*_8_/*K*_7_ = 1.17/0.80), then a small amount of trifluoroacetic acid was added to this solvent system to modify the *K* values of **7** and **8**. Surprisingly, the *K* value of **7** increased dramatically after the solvent system was modified by trifluoroacetic acid, indicating that **7** might be an acid compound. Thus, we decided to separate **7** using pH-zone-refining CCC, which is a technique for the separation of ionizable compounds [23].

A successful pH-zone-refining CCC separation usually relies on the selection of a suitable solvent system that has a satisfactory *K* value under both acidic (*K*_acid_ ≫ 1) and basic (*K*_basic_ ≪ 1) conditions [23]. As shown in Table 4, three solvent systems were modified with 26 mM of trifluoroacetic acid and 45 mM of ammonia solution, respectively, and then, the modified solvent systems were used to determine the *K*_acid_ values and *K*_basic_ values of compounds **7** and **8**. Since the *K*_acid_ value (15.80) and *K*_basic_ value (0.03) that the solvent system *n*-butanol/water (1:1, *v*/*v*) offered were acceptable, the solvent system *n*-butanol/water (1:1, *v*/*v*) was selected for pH-zone-refining CCC separation. Briefly, the solvent system *n*-butanol/water (1:1, *v*/*v*) was first partitioned to upper and lower phases, and then the partitioned upper layer was acidified with 26 mM of trifluoroacetic acid to be used as the stationary phase; while the partitioned lower layer was basified with 45 mM of ammonia solution to be used as the mobile phase.

Then, the separation was carried out as described in Section 3.9.1. As shown in Figure 5A, about 52.5 mg of compound **7** (Figure 5D) was separated from 165 mg of a mixture of **7** and **8** (Figure 5C); however, it was not pure at 300 nm, as detected using another HPLC column (Figure 5F). Then, compound **7** was further purified using preparative HPLC, as shown in Figure 5B. About 10 mg of high-purity compound **7** (Figure 5G) was separated from 50 mg of the impure sample (Figure 5F). Notably, the HPLC peak in Figure 5D at 35 min and the HPLC peak in Figure 5F all belong to compound **7**, but they were determined by two different HPLC columns, as described in Section 3.6.

However, compound **5** was not separated in this study, since it degraded during preservation, as shown in Appendix A.

### 2.5. Structure Identification

A total of eight compounds were separated from the 70% methanol extract of *M. volcanica* and identified via NMR, EI-MS, and LC-MS analysis as well as by comparison with a previously published paper and a standard compound (rutin). These compounds are shikimic acid (**2**) [29], gallic acid (**3**) [30], protocatechuic acid (**4**) [31], 4,4′-dihydroxy-3,3′-imino-di-benzoic acid (**6**) [32], miquelianin (**7**) [33], rutin (**8**) [34], quercitrin (**9**) [35], and quercetin (**10**) [36], as shown in Appendix A and Figure 6. Moreover, the raw NMR spectra are shown in Appendix A. Notably, we report the structural information of these eight compounds from *M. volcanica* for the first time, which is supposed to provide essential phytochemical evidence and promote the research and application of this plant [16].

### 2.6. The Activity of the Separated Compounds

#### 2.6.1. Antioxidant Activity

The evaluation of the separated compounds was carried out using DPPH, ABTS, and ORAC assays. The fluorescein fluorescence decay cure induced by AAPH in the presence of Trolox and the separated compounds and the calibration cures of DPPH and ABTS radical inhibition (%) or AUC value (ORAC assay) by Trolox are presented in Appendix A, respectively, in the Appendix A. The antioxidant activities of the compounds tested were expressed as TEAC values (µM Trolox/µM compound). As shown in Table 5, except for shikimic acid, all the other compounds showed high antioxidant potentials, with the TEAC values ranging from 0.72 to 3.01 against DPPH radicals, from 0.88 to 3.61 against ABTS radicals, and from 1.43 to 6.93 against peroxyl radicals (ORAC assay). Among the compounds tested, gallic acid exhibited the highest antioxidant potential against the DPPH radical with a TEAC value of 3.01, whereas quercetin was found to be the most active compound against ABTS and peroxyl radicals with TEAC values of 3.61 and 6.39, respectively. Moreover, quercetin appeared to have higher activity against DPPH, ABTS, and peroxyl radicals than its 3-*O*-glycosides miquelianin (quercetin 3-*O*-glucuronide), quercitrin (quercetin 3-*O*-rhamnoside), and rutin (quercetin 3-*O*-rutinoside), indicating that the 3-*O*-glycosylation of quercetin may diminish its antioxidant potential, which is in accordance with a previous study carried out by Cai et al. [37]. Notably, in addition to in vitro antioxidant potentials, some of the compounds tested have been proved to be able to ameliorate the oxidative stress in streptozotocin-induced diabetic rats, including gallic acid [38], protocatechuic acid [39], rutin [40], quercitrin [41], and quercetin [42].

#### 2.6.2. Antiglycation Activity

As shown in Table 5, compounds 4,4′-dihydroxy-3,3′-imino-di-benzoic acid, miquelianin, rutin, quercitrin, and quercetin showed considerable inhibitory activities toward the formation of AGEs by the glycation of BSA with fructose, with IC_50_ values ranging from 17.65 to 60.46 µM, compared with that of the positive control aminoguanidine hydrochloride (IC_50_ 2180.9 µM). Among the compounds tested, quercitrin showed the most potent inhibitory activity against the fructose-mediated BSA glycation, which also exhibited considerable antiglycation activity in a previous study [43]. Moreover, since 48.41% (close to 50% inhibition) antiglycation activity was observed for protocatechuic acid at 100 µM, it was supposed to be more active than aminoguanidine (IC_50_ 2180.9 µM) against protein glycation. However, within the concentrations (12.5–100 µM) tested, gallic acid showed a weak antiglycation activity, whereas shikimic acid was inactive. In addition to the inhibition of fructose-mediated BSA glycation, flavonoids including quercitrin [44], rutin [44], and quercetin [45] can also inhibit AGEs by trapping methylglyoxal, which is a highly reactive AGEs precursor implicated in diabetic complications [46]. However, 4,4′-dihydroxy-3,3′-imino-di-benzoic acid and miquelianin are reported to have antiglycation potential for the first time in this study. Although no in vivo studies have been carried out yet to evaluate the antiglycation activities using the compounds reported in this study, the antiglycation potentials of some compounds have been observed in animal studies, including kaempferol [47], phloretin [48], and tea polyphenols [48,49], verifying the beneficial effects of AGEs inhibitors.

#### 2.6.3. Aldose Reductase Inhibitory Activity

As shown in Table 5, quercitrin showed the most potent AR inhibitory activity (IC_50_ 0.17 µM), followed by rutin (IC_50_ 3.51 µM), miquelianin (IC_50_ 9.04 µM), quercetin (IC_50_ 16.57 µM), and 4,4′-dihydroxy-3,3′-imino-di-benzoic acid (IC_50_ 47.00 µM), whereas shikimic acid, gallic acid, and protocatechuic acid showed weak activity against AR within the concentrations tested (6.25–50 µM). Results indicated that the 3-*O*-glycosylation of quercetin with D-glucuronide (miquelianin), D-rutinoside (rutin), and particular L-rhamnose (quercitrin) could increase the AR inhibitory activity of quercetin, which was also observed in previous studies [43,50]. An in vivo study revealed that the oral administration of quercitrin significantly decreased the accumulation of sorbitol in the lens of diabetic Octodon degus and delayed the onset of cataract [51]. Moreover, rutin was able to suppress the accumulation of sorbitol in human erythrocytes under high glucose conditions [50], and the administration of quercetin and rutin could delay the progression of lens opacification in diabetic rats through the inhibition of oxidative stress and the polyol pathway [26]. In addition, rutin and quercetin could partially reverse the impaired cardiac function in streptozotoci*n*-induced diabetic cats possibly through the inhibition of AR [52]. However, compounds 4,4′-dihydroxy-3,3′-imino-di-benzoic acid and miquelianin are reported to have AR inhibitory activities for the first time in this study. Nevertheless, considering that miquelianin exhibited greater AR inhibitory activity than quercetin, miquelianin, similar to quercetin, also has the potential to prevent or treat diabetic complications. Furthermore, in addition to the prevention or treatment of diabetic complications, AR inhibitors also have the potential to suppress oxidative stress-induced inflammatory complications [1,53,54].

In addition, considering that shikimic acid was inactive against free radicals, AGEs, and AR, the anti-diabetic potential of the extract can be further improved by removing shikimic acid using certain methods, e.g., solvent–solvent partition, whereas the byproduct can be a potential alternative source for the industrial production of shikimic acid, which is an important drug precursor in the pharmaceutical industry [55].

## 3. Materials and Methods

### 3.1. Reagents and Materials

The organic solvents used for extraction and CCC separation were of analytical grade (Samchun Pure Chemical Co., Ltd., Pyeongtaek-si, Korea). The methanol used for high-performance liquid chromatography (HPLC) and preparative HPLC was of HPLC grade (J.T.Baker^®^, Avantor Performance Materials, LLC, Center Valley, PA, USA). The water used in this study was ultrapure water generated by a Milli-Q water purification system (Millipore, Bedford, MA, USA). Ammonium sulfate was purchased from Merck (Darmstadt, Hesse, Germany). Quercetin, aminoguanidine hydrochloride, Trolox, DL-glyceraldehyde (dimer), *β*-nicotinamide adenine dinucleotide 2′-phosphate reduced tetrasodium salt hydrate (NADPH), 2,2-diphenyl-1-picrylhydrazyl (DPPH), fluorescein sodium salt, 2,2′-Azobis(2-methylpropionamidine) dihydrochloride (AAPH), sodium azide, fructose, potassium persulfate, 2,2′-Azino-bis(3-ethylbenzothiazoline-6-sulfonic acid) diammonium salt (ABTS), potassium phosphate monobasic, sodium hydroxide, sodium phosphate monobasic dihydrate, sodium phosphate dibasic dodecahydrate, rutin hydrate, dimethyl sulfoxide, trifluoroacetic acid (99%), and ammonia solution (28–30%) were purchased from Sigma-Aldrich Chemical Co. (St. Louis, MO, USA). Bovine serum albumin (BSA) was purchased from Bovogen Biologicals (Bovostar #BSA100; East Keilor, Australia). Notably, all the *n*-butanol used in this study was pre-saturated with water.

The dried leaves of *M. volcanica* were obtained from the department of La Libertad in Peru in a local market and deposited at the Center for Efficacy Assessment and Development of Functional Foods and Drugs, Hallym University as described in our previous study (synonym *Muehlenbeckia volcanica* Meisn.) [14].

### 3.2. Preparation of Plant Extract

The dried leaves of *M. volcanica* (331 g) were extracted twice, each time two days, using 4 L of 70% methanol aqueous solution at room temperature (about 25 °C). Then, the extraction solution was combined, filtrated (5 µm, Advantec #2), and evaporated by rotary evaporation at 37 °C to yield about 71 g of dried extract.

### 3.3. Antioxidant Assay

The antioxidant activities of the 70% methanol extract of *M. volcanica* and the separated compounds were evaluated using DPPH, ABTS, and ORAC assays. The results were expressed as Trolox equivalent antioxidant capacity (TEAC, µg Trolox/µg extract or µM Trolox/µM compound), which were calculated with the equations obtained from the calibration curves of Trolox standards (final concentrations) versus their DPPH and ABTS radical inhibition (%) and the net AUC value (ORAC assay). A larger TEAC value indicates that the sample tested has higher antioxidant activity.

#### 3.3.1. DPPH Radical Scavenging Assay

The DPPH radical scavenging assay was carried out using an EL800 microplate reader (Bio-Tek Instruments, Winooski, VT, USA) [56]. Briefly, 180 μL of freshly prepared DPPH solution (0.32 mM in methanol) was mixed with 20 μL of sample (dissolved in methanol, extract 250 µg/mL; compounds 62.5–1000 µM) in a 96-well plate. The mixture was incubated for 20 min in the dark at 25 °C, and then, the absorbance was measured at 570 nm. Trolox was used as a positive control, which showed a linear radical inhibition curve within final concentrations of 1.56 µg/mL (6.25 µM) and 25 µg/mL (100 µM). Results are expressed in µg Trolox equivalents/µg extract or µM Trolox equivalents/µM compound. The DPPH radical scavenging activity (%) is calculated using Equation (1):(1)$$\%inhibition=\left(1-\frac{A_{sample}-A_{blank1}}{A_{control}-A_{blank2}}\right)\times 100\%,$$
where *A_sample_* is the absorbance of DPPH solution with a sample, *A_blank_*_1_ is the absorbance of the test sample without DPPH, *A_control_* is the absorbance of DPPH solution without a sample, *A_blank_*_2_ is the absorbance of methanol, without DPPH or a sample.

#### 3.3.2. ABTS Radical Scavenging Assay

The ABTS radical scavenging assay was carried out using the same EL800 microplate reader [56]. Briefly, 0.2 mM of ABTS diammonium salt was prepared in 3.5 mM of potassium persulfate aqueous solution, and the mixture solution was then diluted 10 times using distilled water. The diluted solution was allowed to stand for 24 h in the dark at room temperature (about 25 °C) to produce ABTS radical cations (ABTS^+^). For reaction, 290 μL of ABTS^+^ solution was mixed with 10 μL of sample (dissolved in methanol, extract 62.5–125 µg/mL; compounds 20–250 µM) in a 96-well plate, and the mixture solution was allowed to react for 10 min at 25 °C in the dark. Then, the absorbance was measured at 750 nm. Trolox was used as a positive control, which showed a linear radical inhibition curve within a final concentrations of 0.13 µg/mL (0.52 µM) and 4.17 µg/mL (16.67 µM). Results are expressed in µg Trolox equivalents/µg extract or µM Trolox equivalents/µM compound. The ABTS radical scavenging activity (%) is calculated using Equation (1) where *A_sample_* is the absorbance of ABTS^+^ solution with a sample, *A_blank_*_1_ is the absorbance of the test sample without ABTS^+^, *A_control_* is the absorbance of ABTS^+^ solution without a sample, *A_blank_*_2_ is the absorbance of buffer, without ABTS^+^ or a sample.

#### 3.3.3. ORAC Assay

The ORAC assay was carried out using a Fluoroskan Ascent FL microplate reader (Thermo, Waltham, MA, USA) using the method described previously [57,58] with minor modifications. Antioxidants can scavenge the peroxyl radicals generated by AAPH, which prevent the degradation of the fluorescein and, therefore, prevent the loss of the fluorescence of fluorescein. Briefly, fluorescein sodium salt (117 nM) and AAPH (40 mM) were freshly prepared in 0.1 M phosphate-buffered saline (PBS, pH 7.4). For reaction, 120 µL of fluorescein sodium salt (117 nM) and 20 µL of sample (dissolved in methanol, extract 10–20 µg/mL; compounds 10–160 µM) were mixed in a black 96-well plate and pre-incubated for 15 min at 37 °C. Next, 60 µL of AAPH (40 mM) was added, and the fluorescence was measured 100 times (interval 1 min) at excitation and emission wavelengths of 485 nm and 538 nm, respectively, by the microplate reader maintained at 37 °C. Trolox was used as a positive control, which showed a linear net AUC (the area under the curve) values cure within concentrations of 0.50 µg/mL (2.00 µM) and 4.00 µg/mL (16.00 µM), whereas the net AUC was calculated by subtracting the AUC of the blank group (without samples) from that of a sample group. The relative ORAC values are expressed in µg Trolox equivalents/µg extract or µM Trolox equivalents/µM compound. The AUC is calculated using Equation (2):(2)$$\mathrm{AUC}=1+\sum_{n=1}^{n=99}f_{n}/f_{0},$$
where *f*_0_ is the initial fluorescence of the fluorescein reading at 0 min and *f_n_* is the fluorescence reading at time *n*.

### 3.4. Antiglycation Assay

The antiglycation assay was carried out using a fructose-mediated BSA glycation model as reported previously [43] with minor modifications. In brief, 50 mg/mL of BSA was prepared using 0.033% sodium azide-containing 0.1 M PBS (pH 7.4), and 286 mM of fructose was prepared using 0.1 M PBS (pH 7.4). Then, 600 µL of BSA solution (50 mg/mL) was incubated with 350 µL of fructose solution (286 mM) and 50 µL of sample solution (dissolved in 50% methanol aqueous solution, extract 0.25–2 mg/mL; compounds 0.5–2 mM) in the dark at 37 °C for 14 days. Then, 200 µL of the reaction solution from each sample was pipetted to black 96-well plates to determine the fluorescent AGEs using a microplate fluorescence reader (BIOTEK-FLx800, Winooski, VT, USA) at excitation and emission wavelengths of 360 ± 20 nm and 460 ± 20 nm, respectively. Aminoguanidine [25], a synthetic antiglycation agent, was used as a positive control with final concentrations of 200 µg/mL to 800 µg/mL or 1 mM to 4 mM. The antiglycative activity (%) was calculated using Equation (3):(3)$$\%inhibition=\left(1-\frac{f_{sample}-f_{blank1}}{f_{control}-f_{blank2}}\right)\times 100\%,$$
where *f_sample_* is the fluorescence of fructose-mediated AGEs in the presence of a sample; *f_blank_*_1_ is the fluorescence of the mixture solution of BSA and a sample in the absence of fructose; *f_control_* is the fluorescence of fructose-mediated AGEs in the absence of a sample; *f_blank_*_2_ is the fluorescence of BSA without fructose or samples.

### 3.5. Aldose Reductase Inhibition Assay

#### 3.5.1. Preparation of Aldose Reductase

The experimental procedure was approved by the Institutional Animal Care and Use Committees (IACUC) of Hallym University (approval number Hallym-2016-95). The eye lenses of 10-week-old Sprague–Dawley rats (weight 250–280 g) were removed and kept at −70 °C. For the extraction of AR, the rat eye lenses were grounded in a precooled mortar (−70 °C) and extracted using 0.1 M PBS (pH 6.2; about 0.5 mL of PBS per two frozen rat lenses). Then, they were centrifuged at 10,000× *g* for 30 min at 4 °C (Centrifuge 5417R, Eppendorf, Germany) to obtain AR homogenate (in the supernatant) [14].

#### 3.5.2. Aldose Reductase Inhibition Assay

A 96-well plate-based AR inhibition assay was used in this study, which was modified from a previous method [14]. In principle, the rate of the absorbance decrease of NADPH at 340 nm (OD340 nm) was used for evaluation of the AR inhibitory activity. In brief, 110 µL of 0.1 M PBS (pH 7.0), 20 μL of AR homogenate, 20 μL of NADPH (2.4 mM in 0.1 M PBS, pH 8.0), 10 μL of sample (extract 12.5–200 μg/mL; compounds 0.98–1000 μM), and 20 μL of ammonium sulfate solution (4 M in 0.1 M PBS, pH 7.0) were sequentially added into a 96-well plate. Then, as soon as 20 μL of the substrate (DL-glyceraldehyde dimer, 25 mM in 0.1 M PBS, pH 7.0) was added, the absorbance of the reaction solution at 340 nm was measured for 6 min using an Epoch microplate spectrophotometer (BioTek Instruments, Highland Park, IL, USA). Quercetin was used as a positive control. The samples were first prepared at high concentrations in DMSO or in a mixture of DMSO and water and then diluted using water shortly before the experiment. The concentration of DMSO was kept within 0.2% (*v*/*v*) in the reaction system. Particularly, in order to ensure the results obtained from different times of experiments are comparable, the concentration of AR stock homogenate was adjusted by a proper dilution using 0.1 M PBS (pH 6.2) to satisfy the value of |*Slope_c_*| with about 0.038–0.042 as explained below. The AR inhibition ratio (%) for the samples was calculated using Equation (4):(4)$$\%inhibition=\left(1-\frac{\left|Slope_{s}\right|-\left|Slope_{b}\right|}{\left|Slope_{c}\right|-\left|Slope_{b}\right|}\right)\times 100\%,$$
where *Slope_c_*, *Slope_s_*, and *Slope_b_* are the slopes obtained from the OD340 nm versus the reaction time (min) dotted lines of the control group (without sample), sample group (with enzyme and sample), and blank group (without enzyme or sample), respectively. |*Slope*| is the absolute value of Slope.

### 3.6. HPLC Analysis

The HPLC detection of the 70% methanol extract of *M. volcanica*, its fractions, and its compounds was carried out using Dionex equipment (Sunnyvale, CA, USA) equipped with a P850 pump, an ASI-100 automated sample injector, an STH585 column oven (maintained at 30 °C), and a UVD170S detector. The gradient elution of samples (10 µL) was carried out using acidified water (0.1% trifluoroacetic acid, A) and methanol (B) at 0.7 mL/min, as follows: 5–25% B from 0–15 min; 25–60% B from 15–40 min; 60–100% B from 40–43 min; 100–5% B from 43–45 min; and 5% B from 45–55 min. The eluate was monitored at 225, 250, and 300 nm. The samples were mainly analyzed using a Phenomenex Synergi Hydro-RP 80 Å column (150 × 4.60 mm, 4 µm), except the two samples shown in Section 2.4, a low-purity compound **7** (300 nm) and a high-purity compound **7** (300 nm), which were analyzed using an Agilent ZORBAX Eclipse XDB-C18 80 Å column (4.6 × 150 mm, 5 µm).

### 3.7. Separation of Compounds by Polarity-Stepwise Elution CCC

#### 3.7.1. Selection and Preparation of Polarity-Stepwise Elution CCC Solvent Systems

Solvent methanol/water (1:5, *v*/*v*) was selected as the stationary phase for polarity-stepwise elution CCC, while five mobile phases were selected to sequentially elute the compounds from the extract with polarities from low to high, including solvents *n*-hexane/ethyl acetate (3:5, *v*/*v*), *n*-hexane/ethyl acetate (2:5, *v*/*v*), *n*-hexane/ethyl acetate (1:5, *v*/*v*), ethyl acetate, and ethyl acetate/*n*-butanol (4:1, *v*/*v*). The selection of the polarity gradient solvent systems mainly depended on the suitable partition coefficients (*K* value) and settling times the solvent systems offered. The *K* values were calculated as previously described [59]. Each solvent system was paired with equal volumes of solvent methanol/water (1:5, *v*/*v*) and individual *n*-hexane/ethyl acetate (3:5, *v*/*v*), *n*-hexane/ethyl acetate (2:5, *v*/*v*), *n*-hexane/ethyl acetate (1:5, *v*/*v*), ethyl acetate, and ethyl acetate/*n*-butanol (4:1, *v*/*v*) was partitioned to the upper layer and lower layer. Then, a proper amount of sample (1–2 mg, in 1.5 mL tube) was dissolved by equal volumes of the upper and lower phases (each 500 µL) assisted by vortex mixing. After settling, the upper layer and lower layer sample solutions were individually separated (each 200 µL) and then evaporated to dryness by nitrogen gas. Then, the residue was re-dissolved using 200 µL of methanol and subjected to HPLC assay. The *K* value was calculated as *K* = *A_upper_*/*A_lower_*, where *A_upper_* and *A_lower_* were the HPLC peak areas of the compound tested in the upper and lower layers, respectively.

The settling times of each newly paired solvent system, consisting of equal volumes of upper and lower phases (each 3 mL) in a 30 mL glass tube, were checked immediately after vigorous shaking of the solvent system. Additionally, the phase ratio, the upper layer to lower layer, of each newly paired solvent system was also determined. Briefly, equal volumes of upper and lower phases (each 4 mL) were prepared in a 10 mL glass graduated cylinder; then, they were vigorously mixed using a pipette for at least 20 s. After settling, the volume ratio of the upper phase to the lower phase was calculated.

Equal volumes of solvents methanol/water (1:5, *v*/*v*) and *n*-hexane/ethyl acetate (3:5, *v*/*v*), each 1 L, were thoroughly mixed in a separating funnel and separated after settling. The partitioned lower layer, methanol/water (1:5, *v*/*v*), was selected as the stationary phase, and the partitioned upper layer, *n*-hexane/ethyl acetate (3:5, *v*/*v*), was selected as the first step mobile phase, mobile phase I; whereas solvents *n*-hexane/ethyl acetate (2:5, *v*/*v*), *n*-hexane/ethyl acetate (1:5, *v*/*v*), ethyl acetate, and ethyl acetate/*n*-butanol (4:1, *v*/*v*), each 500 mL, were individually prepared in 1 L glass bottles and selected as mobile phase II, mobile phase III, mobile phase IV, and mobile phase V, respectively. The stationary and mobile phases were degassed by sonication for at least 20 min before use.

#### 3.7.2. Separation by Polarity-Gradient Elution CCC

All CCC separations were carried out using a TBE 300C HSCCC system (Tauto Biotech. Co., Ltd., Shanghai, China) with three preparative coils (diameter of tube: 2.6 mm; total volume: 300 mL). An Isolera FLASH purification system (Biotage, Uppsala, Sweden) was equipped with the HSCCC system as a pump, a UV monitor, and an auto fraction collector.

The polarity-stepwise elution CCC separation was conducted using a tail-to-head elution mode. In brief, the CCC coil was first filled with the stationary phase, solvent methanol/water (1:5, *v*/*v*), and then, the rotation speed of the apparatus was adjusted to 900 rpm. Then, the mobile phase I was pumped in at 3 mL/min until a hydrodynamic equilibrium was achieved, as indicated by a steady elution of the mobile phase from the column outlet line. After that, about 2 g of sample was prepared in 17 mL of a solvent system composed of the stationary phase and the mobile phase I and loaded to the sample loop (maximum 20 mL) for separation. Five mobile phases were used to sequentially elute the compounds with polarities from low to high at 3 mL/min as follows: mobile phase I, 350–600 mL; mobile phase II, 600–970 mL; mobile phase III, 970–1340 mL; mobile phase IV, 1340–1480 mL; and mobile phase V, 1480–1890 mL. The eluate was monitored at 225 and 250 nm and automatically collected according to 250 nm. After the separation was completed, the stationary phase was then pumped out by air and collected by a graduated cylinder for calculation of the stationary phase retention ratio as *V_st_*/*V_coil_*, where *V_coil_* is the volume of the CCC coiled column and *V_st_* is the volume of the stationary phase.

### 3.8. Separation and Purification of Compound ***2*** by Conventional HSCCC and Preparative HPLC

#### 3.8.1. Separation of Compound **2** by Conventional HSCCC

The solvent system *n*-butanol/water (1:1, *v*/*v*), 2 L, was acidified with 26 mM of trifluoroacetic acid, thoroughly mixed in a separating funnel, and separated after settling. The separated upper layer and lower layer were degassed by sonication and were used as the stationary phase and mobile phase, respectively, for conventional HSCCC separation.

Compound **2** was concentrated by polarity-stepwise elution CCC as a mixture of compounds **1** and **2**, which was then separated by conventional HSCCC. In brief, the HSCCC coil was first filled with the stationary phase, and the rotation rate of the apparatus was then adjusted to 900 rpm. After that, the mobile phase was introduced at 2 mL/min until a hydrodynamic equilibrium was reached, as indicated by a steady elution of the mobile phase from the column outlet line. Then, about 1 g of the mixture of compounds **1** and **2** was dissolved in 14 mL of diphasic solvent system and loaded to a sample loop (maximum 20 mL) for separation at an elution speed of 2 mL/min. The eluate was monitored at 225 and 250 nm and collected according to 250 nm.

#### 3.8.2. Purification of Compound **2** by Preparative HPLC

Compound **2** separated by conventional HSCCC was found to be impure by HPLC analysis at 225 nm, which was further purified by a preparative HPLC (LC-908; JAI, Japan) using a pre-column (Φ20 × 500 mm; JAIGEL-GS310). Briefly, about 190 mg of the impure compound **2** was dissolved in 1.6 mL of water and loaded to the sample loop (maximum 2 mL) for purification. The water was acidified with 13 mM of trifluoroacetic acid and used as the mobile phase to elute compound **2** at 3 mL/min, with the monitoring wavelength set at 225 nm.

### 3.9. Separation and Purification of Compound ***7*** by pH-Zone-Refining CCC and Preparative HPLC

#### 3.9.1. Separation of Compound **7** by pH-Zone-Refining CCC

Equal volumes of *n*-butanol and water (each 1 L) were thoroughly mixed in a separating funnel and then separated after settling. The partitioned upper layer was acidified by trifluoroacetic acid at a concentration of 26 mM (pH = 1.5) and selected as the stationary phase, while the partitioned lower layer was basified by ammonia solution at a concentration of about 45 mM (pH = 11) and used as the mobile phase. Both the stationary and mobile phases were degassed by sonication for at least 20 min before use.

Compound **7** was concentrated by polarity-stepwise elution CCC as a mixture of compounds **7** and **8**, which was further separated by pH-zone-refining CCC. Briefly, the HSCCC coil was first filled with the stationary phase; then, the rotation speed of the apparatus was adjusted to 900 rpm. About 165 mg of the mixture of compounds **7** and **8** was dissolved in 15 mL of the stationary phase and loaded in the sample loop (maximum 20 mL). Then, the separation was initiated by introducing the mobile phase at 3 mL/min. Notably, the eluate was monitored and automatically collected according to UV absorbance at 250 nm rather than according to pH detection.

#### 3.9.2. Purification of Compound **7** by Preparative HPLC

The compound **7** separated by pH-zone-refining CCC was found to be impure at 300 nm by HPLC analysis, which was further purified using the same preparative HPLC as mentioned above. In brief, about 50 mg of the impure compound **7** was dissolved in 1.5 mL of 40% methanol aqueous solution and loaded to the sample loop (maximum 2 mL) for purification. A 40% methanol aqueous solution was acidified with 13 mM of trifluoroacetic acid and used as the mobile phase to elute compound **7** at 4 mL/min, with the monitoring wavelength set at 300 nm.

### 3.10. Structure Elucidation

Structure identification of the purified compounds was mainly carried out using 600 MHz NMR (Bruker Avance Neo 600 Ultra ShieldTM; Bruker Biospin, Germany), 400 MHz NMR (JNM-ECZ400S/L1; JEOL Ltd., Tokyo, Japan), EI-MS (JEOL JMS-700; JEOL Ltd., Tokyo, Japan), LC/MS (AB Sciex QTrap^®^ 4500; Foster City, CA, USA), high-resolution ESI-MS (SYNAPT G2; Waters, Manchester, UK), and comparisons with references.

### 3.11. Statistical Analysis

All activity assays were conducted with at least three repetitions, with the results given as mean ± standard deviations (SD). The half-maximal inhibitory concentrations (IC_50_ values) of antiglycation agents and AR inhibitors were calculated through either linear regression or logarithmic analysis, which should satisfy each regression coefficient with more than 0.95 (r^2^ > 0.95). The statistical analyses of the bioactivities of the separated compounds were carried out using SPSS software (Version 25; IBM, New York, NY, USA). A significant difference exists when *p* < 0.05 (LSD test).

## 4. Conclusions

In a single run of the polarity-stepwise elution CCC, six high-purity compounds, with a wide range of polarities, and two concentrated CCC fractions were obtained. The two CCC fractions were further separated and purified using conventional HSCCC, pH-zone-refining CCC, and preparative HPLC. It is worth highlighting that these eight compounds, identified as shikimic acid (**2**), gallic acid (**3**), protocatechuic acid (**4**), 4,4′-dihydroxy-3,3′-imino-di-benzoic acid (**6**), miquelianin (**7**), rutin (**8**), quercitrin (**9**), and quercetin (**10**), are reported from *M. volcanica* for the first time. Notably, except for shikimic acid, all other compounds showed anti-diabetic potentials via antioxidant, antiglycation, and aldose reductase inhibition. Our results suggested that the polarity-stepwise elution CCC can be used for the efficient separation or fractionation of compounds with a wide range of polarities from natural products. Moreover, *M. volcanica* and its bioactive compounds are potent anti-diabetic agents and are worth being subjected to further in vivo study to verify its anti-diabetic properties. Additionally, the potential synergistic/antagonistic effects among the tested compounds have yet to be studied.

## Figures and Tables

**Figure 1 molecules-26-00224-f001:**
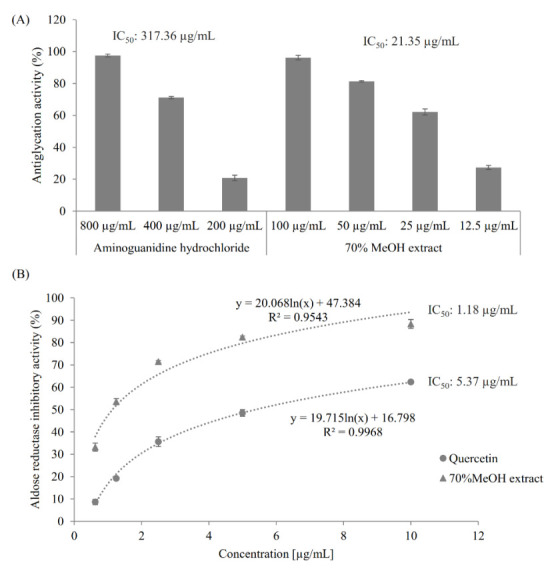
Antiglycation activity and aldose reductase inhibitory activity of the 70% methanol extract of *M. volcanica*. (**A**) Antiglycation activities of the the 70% methanol extract of *M. volcanica* (IC_50_, 21.35 µg/mL) and its positive control aminoguanidine hydrochloride (IC_50_, 317.36 µg/mL). (**B**) Aldose reductase inhibitory activities of the 70% methanol extract of *M. volcanica* (IC_50_, 1.18 µg/mL) and its positive control quercetin (IC_50_, 5.37 µg/mL).

**Figure 2 molecules-26-00224-f002:**
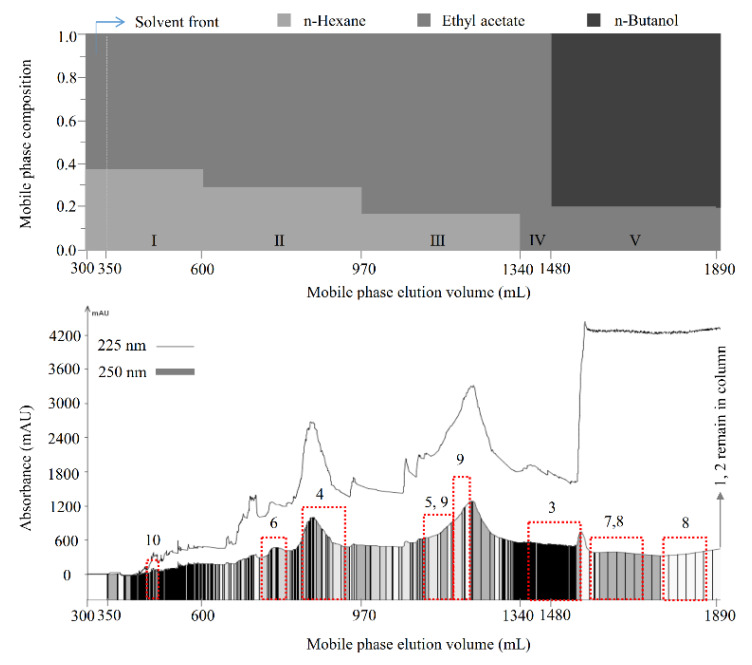
Separation of compounds by polarity-stepwise elution counter-current chromatography (CCC). Solvent methanol/water (1:5, *v*/*v*) was used as the stationary phase. Ten compounds from the 70% methanol extract of *M. volcanica,* with polarities from low to high, were sequentially separated or fractionated by polarity-stepwise elution CCC using five mobile phases at 3 mL/min as follows: mobile phase I, *n*-hexane/ethyl acetate 3:5, *v*/*v*, 350–600 mL; mobile phase II, *n*-hexane/ethyl acetate 2:5, *v*/*v*, 600–970 mL; mobile phase III, *n*-hexane/ethyl acetate 1:5, *v*/*v*, 970–1340 mL; mobile phase IV, ethyl acetate, 1340–1480 mL; mobile phase V, ethyl acetate/*n*-butanol 4:1, *v*/*v*, 1480–1890 mL. The *n*-butanol used was pre-saturated with water in the present study. The eluate was monitored at 225 nm (line) and 250 nm (rectangle) and automatically collected according to 250 nm.

**Figure 3 molecules-26-00224-f003:**
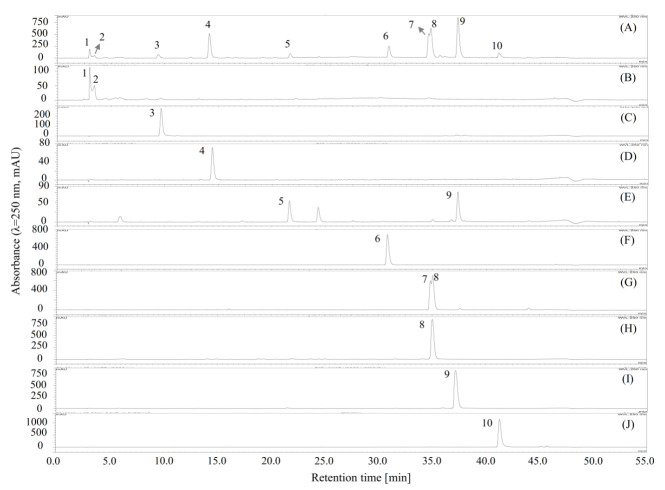
HPLC chromatograms of the separated or fractionated compounds from 70% methanol extract of *M. volcanica* by polarity-stepwise elution CCC. (**A**) HPLC chromatogram of the 70% methanol extract of *M. volcanica*; (**B**–**J**) HPLC chromatograms of the separated or fractionated compounds by polarity-stepwise elution CCC. Moreover, the separated compounds are later identified as shikimic acid (**2**), gallic acid (**3**), protocatechuic acid (**4**), 4,4′-dihydroxy-3,3′-imino-di-benzoic acid (**6**), miquelianin (**7**), rutin (**8**), quercitrin (**9**), and quercetin (**10**).

**Figure 4 molecules-26-00224-f004:**
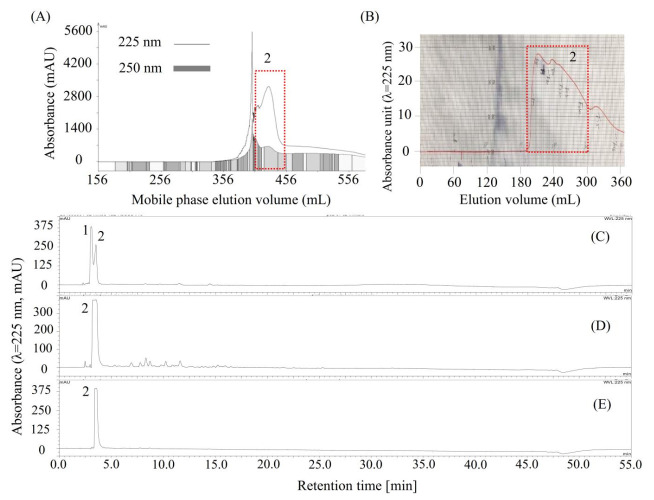
Chromatograms of high-speed counter-current chromatography (HSCCC), preparative HPLC, and analytical HPLC for the separation and purification of compound **2**. (**A**) HSCCC separation of **2** from the mixture of **1** and **2** (**C**). The eluate was monitored at 225 nm (line) and 250 nm (rectangle) and was automatically collected according to 250 nm; (**D**) Compound **2** separated by HSCCC; (**B**) preparative HPLC chromatogram for purification of **2** from impure compound **2** (**D**); (**E**) HPLC chromatogram of the purified **2** by preparative HPLC. Moreover, compound **2** is later identified as shikimic acid.

**Figure 5 molecules-26-00224-f005:**
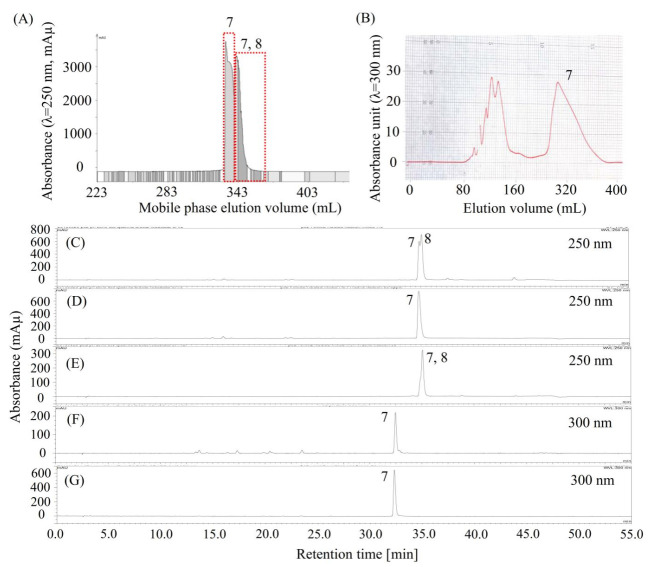
Chromatograms of pH-zone refining CCC, preparative HPLC, and analytical HPLC for the separation and purification of compound **7**. (**A**) Separation of compound **7** by pH-zone refining CCC from the mixture of **7** and **8** (**C**); (**D**,**E**) pH-zone refining CCC fractions of **7** (**D**), and a mixture of **7** and **8** (**E**); (**B**) preparative HPLC chromatogram for the purification of compound **7** from impure compound **7** (**F**); (**G**) purified **7** by preparative HPLC. Notably, chromatograms D and F were obtained from the same compound but analyzed using two different HPLC columns and monitored at different wavelengths. Moreover, compound **7** is later identified as miquelianin.

**Figure 6 molecules-26-00224-f006:**
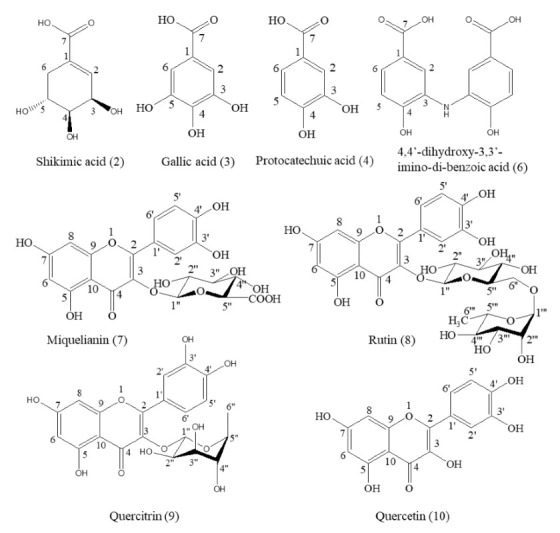
Structures of the compounds separated from the 70% methanol extract of *M. volcanica.*

**Table 1 molecules-26-00224-t001:** Trolox equivalent antioxidant capacity (TEAC) of the 70% methanol extract of *Muehlenbeckia volcanica* (Benth.) Endl.

Antioxidant Assay	Calibration Curves of DPPH and ABTS Radical Inhibition (%) or Net AUC by Trolox ^a^			TEAC ^b^ of the Extract	
	Linearity Range of Trolox (µg/mL)	Progression	r^2^	Concentration (µg/mL) and Radical Scavenging Activity (%) or the Net AUC Value of the Extract	TEAC (µg Trolox/µg Extract)
DPPH	1.56–25.00	y = 2.8123x + 2.9182	0.9999	25.00 (36.53 ± 0.47 ^c^)	0.48 ± 0.01
ABTS	0.13–4.17	y = 23.331x − 1.0452	0.9998	2.08 (23.48 ± 0.41), 4.17 (44.64 ± 0.67)	0.49 ± 0.02
ORAC	0.50–4.00	y = 20.087x + 19.067	0.9939	1.00 (40.19 ± 1.10), 2.00 (65.31 ± 1.92)	1.10 ± 0.07

^a^ The calibration cures were created by plotting DPPH (2,2-diphenyl-1-picrylhydrazyl) and ABTS (2,2′-Azino-bis(3-ethylbenzothiazoline-6-sulfonic acid) diammonium salt) radical inhibition (%) or net area under the curve (AUC) value (ORAC (oxygen radical absorbance capacity) assay) against Trolox concentrations. ^b^ Trolox equivalent antioxidant capacity (TEAC). ^c^ DPPH and ABTS radical inhibition (%), the net AUC value (ORAC assay), and the TEAC value of the extract were given as mean ± standard deviation.

**Table 2 molecules-26-00224-t002:** Partition coefficients (*K*_upper/lower_) of compounds **1**–**10** from the 70% methanol extract of *M. volcanica* offered by the newly paired solvent systems with one stationary phase and individual mobile phases.

HSCCC Mobile Phase Composition ^a^	*K* Values of Compounds 1–10										Settling Times (s)	Phase Ratio ^b^
	1	2	3	4	5	6	7	8	9	10		
***n*-Hexane/ethyl acetate 3:5 ^c^, *v*/*v***	≪1	≪1	0.07	0.51	0.10	0.71	≪1	≪1	0.16	**12.21**	25	0.95
***n*-Hexane/ethyl acetate 2:5 ^d^, *v*/*v***	≪1	≪1	0.15	**0.88**	0.29	**1.49**	0.03	0.03	0.41	15.59	18	1.03
***n*-Hexane/ethyl acetate 1:5 ^e^, *v*/*v***	≪1	≪1	0.38	1.66	0.86	3.18	0.09	0.08	**1.13**	21.81	16	0.91
**Ethyl acetate *^f^***	≪1	≪1	**1.38**	4.14	2.82	7.91	0.46	0.43	3.97	≫1	8	1.00
Ethyl acetate/*n*-butanol 6:1, *v*/*v*	0.10	0.12	2.05	5.29	3.49	10.74	0.80	1.04	7.43	≫1	13	1.03
Ethyl acetate/*n*-butanol 5:1, *v*/*v*	0.18	0.14	2.08	5.19	3.50	12.26	0.90	1.16	7.23	≫1	14	1.05
**Ethyl acetate/*n*-butanol *^g^* 4:1 ^f^, *v*/*v***	0.20	0.09	1.76	4.85	3.04	9.82	**0.80**	**1.17**	7.19	≫1	17	1.11

^a^ Solvent methanol/water (1:5, *v*/*v*) was used as the stationary phase for polarity-stepwise elution HSCCC. ^b^ The phase ratio was calculated as the volume of upper layer divided by that of lower layer. ^c,d,e,f,g^ These five mobile phases (marked in bold) were selected to separate compounds as follows: separation of **10** by *n*-hexane/ethyl acetate (3:5, *v*/*v*); separation of **4** and **6** by *n*-hexane/ethyl acetate (2:5, *v*/*v*); separation of **5** and **9** by *n*-hexane/ethyl acetate (1:5, *v*/*v*); separation of **3** by ethyl acetate; and separation of **7** and **8** by ethyl acetate/*n*-butanol (4:1, *v*/*v*). The *n*-butanol used was pre-saturated with water.

**Table 3 molecules-26-00224-t003:** Partition coefficients (*K*_upper/lower_) of compounds **1** and **2**, and the separation factors (α) between *K*_1_ and *K*_2_ at each solvent system.

Solvent System	*K* _upper/lower_		α*_k_*_1/*k*2_
	1	2	
*n*-Butnaol ^a^/(Methanol/Water 1:5, *v*/*v*) 1:1, *v*/*v*	0.39	0.40	0.98
*n*-Butnaol/(Methanol/Water 1:5, *v*/*v*) 1:1, *v*/*v* + 26 mM TFA ^b^	0.68	0.53	1.29
*n*-Butnaol/Water 1:1, *v*/*v*	0.13	0.28	2.16
*n*-Butnaol/Water 1:1, *v*/*v* + 26 mM TFA	0.59	0.39	1.52

^a^*n*-Butanol was pre-saturated with water. ^b^ Abbreviation of trifluoroacetic acid.

**Table 4 molecules-26-00224-t004:** Partition coefficients (*K*_upper/lower_) of compounds **7** and **8** in different solvent systems under acidic and basic conditions.

Solvent System	*K* _upper/lower_	Compound 7	Compound 8
(Ethyl acetate/*n*-butanol ^a^ 4:1, *v*/*v*)/(methanol/water 1:5, *v*/*v*) 1:1, *v*/*v*	*K* _acid_ ^b^	6.66	1.49
	*K* _base_ ^c^	0.02	0.04
*n*-Butanol/(methanol/water 1:5, *v*/*v*) 1:1, *v*/*v*	*K* _acid_	5.71	3.38
	*K* _base_	0.10	0.21
*n*-Butanol/water 1:1, *v*/*v*	*K* _acid_	15.80	6.51
	*K* _base_	0.03	0.07

^a^*n*-Butanol was pre-saturated with water. ^b^ 26 mM of trifluoroacetic acid was added to each solvent system for determining *K*_acid_ values. ^c^ 45 mM of ammonia solution was added to each solvent system for determining *K*_base_ values.

**Table 5 molecules-26-00224-t005:** Antioxidant, antiglycation, and aldose reductase (AR) inhibitory activities of the compounds separated from the 70% methanol extract of *M. volcanica*.

Sample	TEAC ^1^ (µM trolox/µM Compound)			Antiglycation Activity (%)			AR inhibition Activity (%)		
	DPPH	ABTS	ORAC	Con. ^2^ (µM)	Inhibition (%)	IC_50_ ^3^ (µM)	Con. (µM)	Inhibition (%)	IC_50_ (µM)
Shikimic acid (2)	- ^4^	-	-	100	-	-	50	7.36 ± 0.30	-
				50	-		25	5.86 ± 5.42	
				25	-		12.5	0.32 ± 1.21	
				12.5	-		6.25	-	
Gallic acid (3)	3.01 ± 0.04 ^a^	3.35 ± 0.06 ^b^	1.43 ± 0.08 ^d^	100	8.42 ± 1.27	-	50	13.54 ± 1.21	-
				50	3.69 ± 2.97		25	4.16 ± 1.81	
				25	-		12.5	4.16 ± 5.43	
				12.5	-		6.25	-	
Protocatechuic acid (4)	0.72 ± 0.01 ^f^	0.88 ± 0.03 ^g^	5.79 ± 0.32 ^b^	100	48.41 ± 1.76	-	50	7.78 ± 1.51	-
				50	36.14 ± 0.78		25	7.14 ± 0.60	
				25	11.44 ± 3.92		12.5	2.67 ± 2.71	
				12.5	5.26 ± 4.29		6.25	0.96 ± 2.11	
4,4′-dihydroxy-3,3′-imino-di-benzoic acid (6)	1.38 ± 0.05 ^d^	1.27 ± 0.03 ^c^	5.75 ± 0.26 ^b^	100	68.39 ± 0.92	60.46 ± 2.62 ^b^	50	53.28 ± 2.76	47.03 ± 2.28 ^a^
				50	44.48 ± 2.54		25	27.16 ± 0.36	
				25	14.95 ± 3.87		12.5	14.44 ± 0.60	
				12.5	-		6.25	8.57 ± 0.06	
Miquelianin (7)	1.97 ± 0.02 ^c^	0.99 ± 0.05 ^f^	5.85 ± 0.20 ^b^	100	90.03 ± 1.92	24.51 ± 0.52 ^d^	25	76.65 ± 1.21	7.93 ± 0.40 ^c^
				50	77.99 ± 2.28		12.5	59.13 ± 6.86	
				25	49.38 ± 1.26		6.25	44.46 ± 2.11	
				12.5	24.87 ± 2.12		3.125	29.74 ± 1.21	
Rutin (8)	0.90 ± 0.09 ^e^	1.12 ± 0.04 ^d^	5.77 ± 0.08 ^b^	100	91.51 ± 0.40	20.14 ± 0.23 ^d^	25	84.54 ± 1.51	3.51 ± 0.12 ^d^
				50	81.82 ± 0.65		12.5	75.16 ± 0.90	
				25	61.7 ± 2.49		6.25	62.79 ± 0.30	
				12.5	30.78 ± 1.10		3.125	46.38 ± 1.21	
Quercitrin (9)	0.92 ± 0.01 ^e^	1.07 ± 0.05 ^e^	5.32 ± 0.29 ^c^	100	94.28 ± 1.39	17.65 ± 0.28 ^e^	3.125	94.35 ± 0.30	0.17 ± 0.00 ^e^
				50	89.11 ± 1.26		0.195	54.57 ± 1.78	
				25	75.22 ± 2.52		0.098	33.29 ± 2.17	
				12.5	32.3 ± 0.24		0.049	12.01 ± 0.82	
Quercetin (10)	2.14 ± 0.03 ^b^	3.61 ± 0.03 ^a^	6.93 ± 0.30 ^a^	100	84.31 ± 1.39	36.37 ± 0.13 ^c^	25	60.02 ± 1.21	16.57 ± 0.14 ^b^
				50	67.47 ± 0.24		12.5	42.96 ± 3.02	
				25	35.39 ± 0.92		6.25	26.33 ± 1.21	
				12.5	19.43 ± 1.76		3.125	12.90 ± 2.11	
Aminoguanidine hydrochloride ^5^	-	-	-	4000	81.26 ± 0.88	2180.90 ± 8.11 ^a^	-	-	-
				2000	44.58 ± 0.54		-	-	
				1000	10.99 ± 0.07		-	-	

^1^ Trolox equivalent antioxidant capacity (TEAC). ^2^ Abbreviation of concentration. ^3^ The half-maximal inhibitory concentration (IC_50_). ^4^ Data was given as mean ± standard deviation (SD). Different letters (a, b, c, d, e, f) in each column indicate significant differences (*p* < 0.05); while “-” means inactive or the sample was not determined. ^5^ Positive control for antiglycation assay.

## Data Availability

Data is contained within the article or Appendix A.

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
