# Peer review of "Efficient Separation of Phytochemicals from *Muehlenbeckia volcanica* (Benth.) Endl. by Polarity-Stepwise Elution Counter-Current Chromatography and Their Antioxidant, Antiglycation, and Aldose Reductase Inhibition Potentials"

_molecules, 2021, doi:10.3390/molecules26010224_

Round 1

Reviewer 1 Report

In this manuscript, Zuo et al. have proposed an efficient separation mthods of phytochemicals from Muehlenbeckia volcanica (Benth.) Endl. by polarity stepwise elution counter-current chromatography and have evaluated their antioxidant, antiglycation, and aldose reductase inhibition potentials. The authors utilized appropriate technique of characterization and analysis, supplying useful information. Overall the work is well written and organized, although there are some typing errors to correct during the revision of the work and some methodogical data are missing. Moreover some sentences are too complex and rambling.

 The changes and suggestions are listed below:

  1. Introduction: it is not well organized. The authors start with: “Type 2 diabetes is a metabolic disorder characterized………” . They article doesn’t have as a principal aim the Type 2 diabetes so is necessary change the position of the first 14 line and start with: “Natural product……” Moreover this section is not of fundamental importance so can be significantly shortened.
  2. In the introduction the authors reported: “Recently, it has been 66 reported to have anti-radicals, AR inhibition, and hypoglycemic activities [13,14]” so if this results is already available why they have re-analyzed the same properties of the methonolic extract? The authors have to supply clarification, eventual difference with this already published data or to remove them because the data is alredy present in the literature. I’ m in agreement with the analysis of these properties in the separate compounds but in the methanol extract it is already known.
  1. Please increase the quality of figure 1. It is hard to read what is reported on the axys.
  1. Please indicate in the legend of the figure 3 and 5 what compounds represent the number present in the chromatograms.
  2. Page 13 line 404-05. “Briefly, 180 μL of freshly prepared DPPH solution (0.32 mM 404 in methanol) was mixed with 20 μL of sample (suitable concentrations in methanol) in a 96-well plate”. If the samples have been purified with different solvent the authors are sure they are completely soluble in methanol? It it possible thata tha authors have do an errors and whant to say “…….mixed with 20 μL of sample (dissolved in the suitable solvent)……” Otherwise it is right for the extract but not for the purified compounds whose solubility in methanol is different.
  3. The same above observation have to be consider also for ABTS and ORAC assay, Antiglycation assay and aldose reductase inhibition.
  4. The journal reported in the reference 13 is wrong, please change. Several other errors are present in the references please check.
  5. Page 15 line 476. In “Aldose reductase inhibition assay” there are several errors. In the reported references (13) the authors of the work have utilized DL-glyceraldehyde as substarte not DL-glyceraldehyde dimer as substate. The authors have to clarify otherwise all the obtained results are wrong.
  6. Revise overall the manuscript for the presence of some typing and grammar errors.

Reviewer 2 Report

The article describes isolation of several natural products from Muehlenbeckia volcanica, which is a plant used in traditional medicine in South America. The compounds were also evaluated for its anti-oxidant antiglycation, and aldose reductase 5 inhibition activity. 

The article is technically well done, experiments support the conclusion and the supporting info file contains all important spectral data. 

I can see the novelty of the article in the fact that the plant has not been evaluated for the compound content yet, however, other methods are standard procedures used in the isolation and characterization of natural compounds. 

I only have one question to discuss, how are the activities of the extract before the separation? Does it correspond to the compound content or do you see some synergistic/antagonistic effect? Maybe add one or two sentences to the conclusion part?

oAnyways, this is overall a good paper that deserves to be published, 

Round 2

Reviewer 1 Report

The authors have performed all the requested changes and the work is ready for pubbication.